# Primed Track, high-fidelity lineage tracing in mouse pre-implantation embryos using primed conversion of photoconvertible proteins

Maaike Welling[1,2†], Manuel Alexander Mohr[1,3†‡], Aaron Ponti[1], Lluc Rullan Sabater[1§], Andrea Boni[4#], Yumiko K Kawamura[4], Prisca Liberali[4], Antoine HFM Peters[4,5], Pawel Pelczar[6], Periklis Pantazis[1,2*]

[1]Department for Biosystems Science and Engineering (D-BSSE), ETH Zurich, Basel, Switzerland; [2]Department of Bioengineering, Imperial College London, London, United Kingdom; [3]Howard Hughes Medical Institute, Janelia Research Campus, Ashburn, United States; [4]Friedrich Miescher Institute for Biomedical Research (FMI), Basel, Switzerland; [5]Faculty of Sciences, University of Basel, Basel, Switzerland; [6]Center for Transgenic Models (CTM), University of Basel, Basel, Switzerland

*For correspondence:
p.pantazis@imperial.ac.uk

†These authors contributed equally to this work

Present address: ‡Department of Biology, Stanford University, Stanford, United States; §QuintilesIMS, Basel, Switzerland; #Viventis Microscopy Sàrl, Lausanne, Switzerland

**Abstract** Accurate lineage reconstruction of mammalian pre-implantation development is essential for inferring the earliest cell fate decisions. Lineage tracing using global fluorescence labeling techniques is complicated by increasing cell density and rapid embryo rotation, which hampers automatic alignment and accurate cell tracking of obtained four-dimensional imaging data sets. Here, we exploit the advantageous properties of primed convertible fluorescent proteins (pr-pcFPs) to simultaneously visualize the global green and the photoconverted red population in order to minimize tracking uncertainties over prolonged time windows. Confined primed conversion of H2B-pr-mEosFP-labeled nuclei combined with light-sheet imaging greatly facilitates segmentation, classification, and tracking of individual nuclei from the 4-cell stage up to the blastocyst. Using green and red labels as fiducial markers, we computationally correct for rotational and translational drift, reduce overall data size, and accomplish high-fidelity lineage tracing even for increased imaging time intervals – addressing major concerns in the field of volumetric embryo imaging.
DOI: https://doi.org/10.7554/eLife.44491.001

## Introduction

Accurate lineage tracing and precise tracking of single cells in pre-implantation embryos are essential for a mechanistic understanding of the first cell fate decisions during mammalian development (*Welling et al., 2016*; *Pantazis and Bollenbach, 2012*). Selective plane illumination microscopy (SPIM) has the potential to play a major role in achieving comprehensive, non-invasive imaging of mammalian pre-implantation development. During these early steps of development, a major fraction of embryos (n = 9/19, 45% in this study) exhibit confounding rotational and spatial drift (*Videos 1*, *2* and *3*), which often leads researchers to exclude these embryos from analysis, drastically decreasing efficiency, losing valuable data, and potentially biasing downstream results (*Strnad et al., 2016*; *Motosugi et al., 2005*). While high-imaging rates have helped to overcome these challenges for samples like zebrafish embryos, they demand increased data storage capacities. Moreover, higher frame rates increase photodamage from laser overexposure and are hence less applicable for highly sensitive mouse embryos (*Strnad et al., 2016*; *Takenaka et al., 2007*).

**eLife digest** A mouse embryo starts with one cell, which divides to create identical daughters that quickly start to multiply. Within three to four days, certain cells begin to specialize and take on specific roles. Scientists want to track these early events to understand how they give rise to an individual formed of huge numbers of cells organized in specialized tissues. To do so, researchers genetically manipulate embryos so that each cell produces fluorescent molecules that 'glow' under light. These embryos are grown inside a special microscope for several days. Images are taken regularly and then processed by specialized software that automatically tracks the fluorescent cells and their daughters over time. This helps reconstruct the history of each cell, and which structures they give rise to.

However, many embryos move and turn around between images, and so software packages often lose track of which cell was which. Taking images more frequently is not possible because each imaging event exposes the embryo to light, which can damage its fragile cells.

To address this problem, Welling, Mohr et al. made embryonic cells produce a special fluorescent marker, which is normally green but can be converted to red. Then, a technique known as primed conversion was used so that only one cell in a four-cell embryo would glow red. Welling, Mohr et al. designed a piece of software, baptized 'primed Track', that can use this red cell (and its daughters) to reorient the embryo during image analysis and reliably identify and match any mother cell to its daughters. The new approach means the experiments require fewer imaging events, but also fewer embryos because even the ones that move a lot can be studied. This should help scientists look into how early life processes give rise to specialized cells, and even explore the fate of cells in other tissues.

DOI: https://doi.org/10.7554/eLife.44491.002

Labeling strategies using green-to-red photoconvertible fluorescent proteins (pcFPs) allow for visualization of both the entire population of cells in green and a selected population in red. This combination of global and sparse labeling yields great potential for facilitating lineage tracing and trophectoderm (TE) and inner-cell-mass (ICM) fate assignments after photoconversion (*Kurotaki et al., 2007*). However, to our knowledge these sparse labels have not been combined with SPIM - presumably because photoconversion has been limited by the need for axially unconfined, potentially photodamaging, intense violet light (*Post et al., 2005*). Our recent report of a novel photochemical mechanism called "primed conversion" overcomes this long-standing problem by using dual-wavelength illumination with blue 488nm and far-red 730nm laser light instead (see *Mohr and Pantazis, 2018*) for a review). Importantly, primed conversion allows for confined photoconversion of small volumes in three dimensions (3D) by selectively intersecting the two laser beams in a common focal spot, yielding axial confinement unachievable by 405 nm photoconversion (*Dempsey et al., 2015*; *Mohr et al., 2016*). The discovery of the mechanism responsible for primed conversion enabled the rational engineering of primed convertible ("pr-") variants of most pcFPs (*Mohr et al., 2017*; *Turkowyd et al., 2017*) with improved brightness and photostability, essential properties for long-term imaging in a SPIM (*Mohr et al., 2017*).

Here, we show that primed conversion of single pr-pcFP-labeled cells in early stages of mouse development allows for computational correction of spatial and rotational drift, which minimizes uncertainties in tracking and lineage

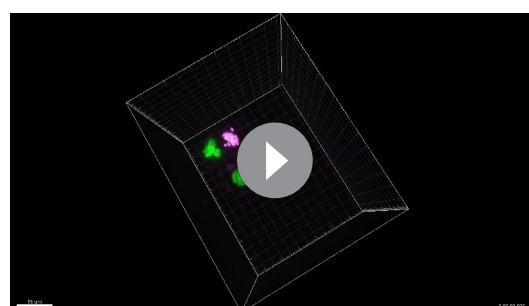

**Video 1.** Video of a developing embryo before drift correction imaged every 15 mintes. Timelapse video of an example embryo, which shows strong spatial and rotational drift before drift correction. pr-mEosFP fluorescence (green) and primed converted pr-mEosFP fluorescence (red). Scale bars, 15 μm; framerate: one frame every 15 min.

DOI: https://doi.org/10.7554/eLife.44491.003

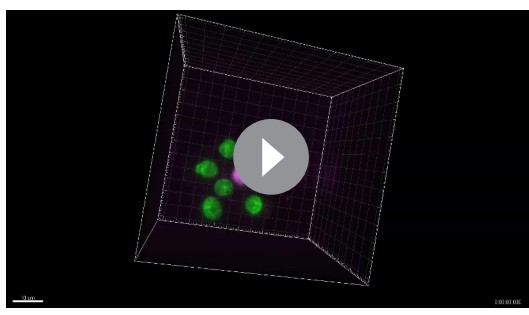

**Video 2.** Video of another developing embryo before drift correction images every 7.5 min. Timelapse video of an example embryo, which shows strong spatial and rotational drift before drift correction. pr-mEosFP fluorescence (green) and primed converted pr-mEosFP fluorescence (red). Scale bars, 10 μm; framerate: one frame every 7.5 min.
DOI: https://doi.org/10.7554/eLife.44491.004

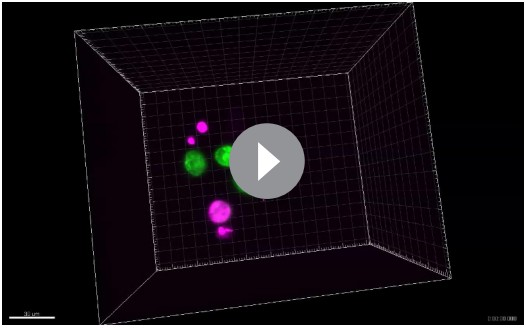

**Video 3.** Video of another developing embryo before drift correction images every 5 min. Timelapse video of an example embryo, which shows strong spatial and rotational drift before drift correction. pr-mEosFP fluorescence (green) and primed converted pr-mEosFP fluorescence (red). Scale bars, 30 μm; framerate: one frame every 5 min.
DOI: https://doi.org/10.7554/eLife.44491.005

tracing. Accurate tracking is achievable even for larger imaging intervals further reducing laser exposure to the sensitive specimen.

## Results and discussion

### H2B-pr-mEosFP-labeled cells primed converted at the 4-cell stage can be visualized up to the blastocyst stage

Previously, we and others found that pr-pcFP variants based on the Eos-family of *Anthozoa*-derived pcFPs efficiently undergo primed conversion and exhibit high levels of photostability and brightness (*Mohr et al., 2017*; *Turkowyd et al., 2017*). In order to assess which protein of the Eos-family is most suitable for long-term cell tracking and lineage tracing experiments in mouse embryos, we directly compared pr-mEos2 and pr-mEosFP. We injected mouse zygotes with mRNAs encoding for the histone fusions H2B-pr-mEos2 or H2B-pr-mEosFP and imaged them at different stages to observe their developmental progression. Embryos injected with mRNA encoding for H2B-pr-mEosFP showed no visible signs of developmental impairment, similar to un-injected control embryos (*Figure 1—figure supplement 1a and 1b*). In contrast, H2B-pr-mEos2-injected embryos showed partly divided, seemingly connected nuclei and prematurely arrested in development (n=30/30) (*Figure 1—figure supplement 1b*). This apparent inability to separate the nuclei during cell division is likely due to a residual tendency of mEos2 to oligomerize, as proposed previously (*Zhang et al., 2012*). As a consequence, we identified primed convertible mEosFP (pr-mEosFP) as the optimal fluorescent protein variant for in vivo primed conversion in the mouse embryo followed by long-term imaging.

Next, we investigated whether a single round of green-to-red photoconversion at the four-cell stage would create a sufficiently large pool of red-converted protein that could be followed throughout development until the blastocyst stage. For this purpose, we performed confined primed conversion in a confocal system as previously described (*Mohr et al., 2016*) to photoconvert a single nucleus of an H2B-pr-mEosFP expressing embryo at the four-cell stage. Primed converted embryos were then transferred and monitored for 60 hours during early embryo development in a custom built SPIM suitable for long term imaging of mouse embryos (*Figure 1a*). To compensate for signal dilution of the H2B-pr-mEosFP signal over time primarily due to cell division, the laser power was gradually increased throughout the imaging sessions. Embryos subjected to photoconversion of a single cell developed normally and the red daughter cells of the initially primed converted cell were clearly distinguishable from non-converted green cells up to the blastocyst stage (*Figure 1b*; *Figure 1—figure supplement 2a*). In addition, primed conversion itself did not impede the

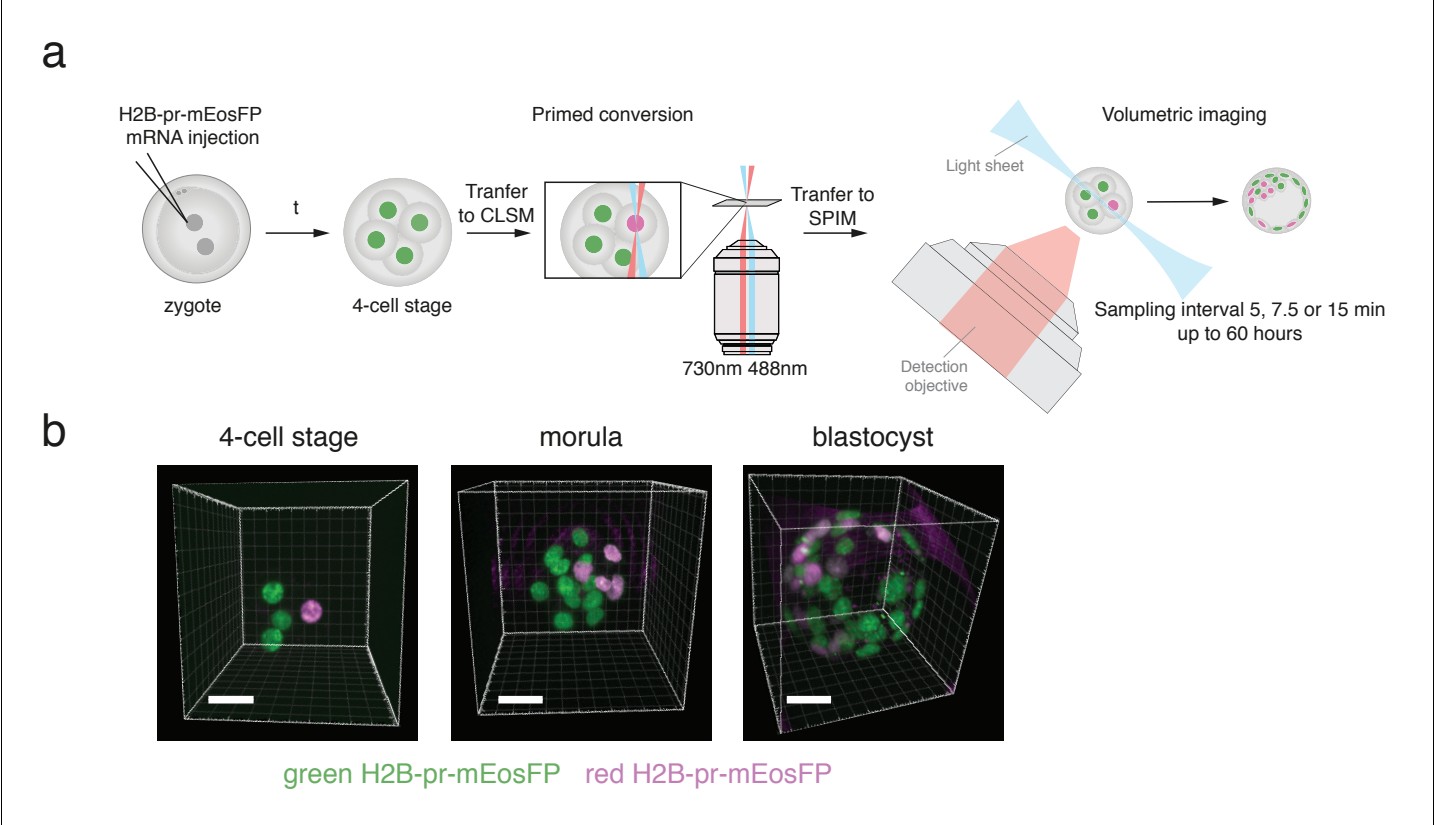

**Figure 1.** H2B-pr-mEosFP injected embryos develop to the blastocyst stage. (a) Experimental setup: Zygotes are injected with H2B-pr-mEosFP mRNA. At the 4-cell stage confined primed conversion of a single nucleus is performed using intersecting 488 nm and 730 nm lasers. The embryos are transferred to an inverted SPIM for non-invasive imaging of their development up to the blastocyst stage. Images are taken every 5, 7.5 or 15 min. (b) Embryos injected with mRNA encoding H2B-pr-mEosFP and converted at the four-cell-stage develop normally and maintain visibility of the red label up to the early blastocyst stage. pr-mEosFP fluorescence (green) and primed converted pr-mEosFP fluorescence (magenta). N $\geq$ 200 embryos out of $\geq$10 independent experiments. Scale bar, 20 μm.

DOI: https://doi.org/10.7554/eLife.44491.006

The following figure supplements are available for figure 1:

**Figure supplement 1.** Embryos expressing H2B-pr-mEosFP develop normally.

DOI: https://doi.org/10.7554/eLife.44491.007

**Figure supplement 2.** Visualizing the lineage of a single cell up to the blastocyst stage using primed converted at the 4-cell stage.

DOI: https://doi.org/10.7554/eLife.44491.008

development of photoconverted embryos compared to non-converted control embryos (*Figure 1— figure supplement 2b*).

## Dual labeling of pre-implantation embryos greatly facilitates automatic segmentation, tracking, and lineage tracing

As cells converted at the four-cell stage can be visualized up to the blastocyst stage, we wondered whether such sparsely labeled subsets of cells could aid computational reorientation and automated lineage tracing in embryos that exhibit dramatic spatial and rotational drift (*Videos 1*, *2* and *3*). Of note, while we initially imaged our embryos with time intervals of 7.5 or 15 minutes, we found that increased sampling frequency did not recover successful lineage tracing for rotating embryos: the percentage of embryos showing spatial and rotational drift prohibitive of automated lineage tracing in our experiments with 5-minute imaging time intervals (=50%, n=8) was similar to those imaged with larger time intervals (=45% embryos imaged every 7.5 or 15 minutes, n=11). To accomplish accurate tracking, we developed a computational pipeline, referred to as "primed Track", for automated segmentation, cell tracking, and lineage tracing. Primed Track uniquely takes advantage of

the sparse red cell population to correct for spatial and rotational drift as well as to simplify lineage reconstruction (*Figure 2a*). In the 5-dimensional (5D, that is 3 spatial dimensions, time, color) imaging data, cells were first segmented based on size, shape, and fluorescence taking into account both color channels. The use of increasing laser power to compensate for red signal dilution mainly due cell division resulted however in increasing background fluorescence. To discern red signal from increasing background signal, we took advantage of the dual nuclear labeling that allowed us to identify weaker fluorescent red nuclei at advanced time points by their overlap with the green signal in which lower autofluorescence was detected (*Figure 2a*, left column). Background signal that was falsely segmented in the green channel due to increased illumination could be excluded by ignoring spots detected outside of a defined radius of the embryo. The ability to select parameters that match the brightness, size, and shape of the embryos combined with fluorescence information of two channels makes the segmentation both robust and flexible for use in different experiments. In addition, the dual color information allowed for cell distinction in instances otherwise rendered ambiguous through high cell density and proximity of nuclei. For instance, we were able to distinguish nuclei that would have been identified as a single nucleus even after manual validation (*Figure 2—figure supplement 1a-c*).

In a second step, the embryo was positioned at its fluorescence center of mass, cropped and rotated, such that the red center of mass was oriented to the same side of the embryo in every time frame to compensate for rotational and spatial drift (*Figure 2a*, middle column; *Videos 4*, *5* and *6*). The resulting high-quality 5D cropped and registered datasets were reduced to only 34±11% of the original size (*Figure 2—figure supplement 2*). The automatic tracking of a realigned embryo resulted in greatly improved lineage tracing fidelity compared to a naïve state-of-the-art lineage-tracing algorithm that was not able to reconstruct a lineage tree from rotating and spatially drifting embryos imaged with a time interval of 15, 7.5 or 5 minutes (Bitplane Imaris cell lineage package) (*Figure 2b*; *Figure 2—figure supplement 2*). Of note, none of the existing state-of-the-art lineage tracing tools such as Ilastik, TrackMate and the TGMM software (*Amat et al., 2014*) were designed to compensate for heavy rotational and spatial drift and are therefore incapable of calculating lineage trees from these embryos (*Figure 2—source data 1*). Separating the green and red channels to generate two less complex datasets during lineage reconstruction further increased the fidelity of lineage tracing versus a dataset consisting of the green channel alone (*Figure 2a*, right column; *Figure 2—figure supplement 3*). We assessed the power of primed Track by comparing the lineage trees obtained i) without corrections, ii) after embryo realignment with all algorithmic corrections, and iii) after final manual review by calculating the total distance between these lineage trees (see Materials and methods for more details) (*Zhang and Shasha, 1989*). We were able to recover all rotating embryos that we acquired using this image analysis pipeline and the resulting lineage trees required a minimal amount of time for manual corrections (0.5-1.5 hours per lineage tree).

## Primed Track allows for decreased sampling frequency

The observation that the registration of embryos based on the dual labeling with primed Track allows for reliable lineage tracing despite heavy embryo rotation suggests that decreasing the imaging frequency will have limited effect on lineage tracing ability. To test the robustness of primed Track, we removed time points from datasets from both rotating as well as non-rotating embryos to examine lineage tracing capacity at a decreased sampling rate. Naturally, larger imaging time intervals increase embryo displacement in consecutive time points and exacerbate the accurateness of lineage tracing (*Figure 3*). Using primed Track to correct for spatial and rotational drift results in reliable reconstruction of the lineage trees even at imaging time intervals of 30 minutes for a rotating embryo and 40 minutes for a non-rotating embryo (*Figure 3*; *Figure 3—figure supplement 1*). While the registration of non-drifting embryos does not increase lineage tracing accuracy for high 5-minute sampling rates, the gain of fidelity in lineage tree reconstruction of these embryos greatly benefits from our presented approach when the imaging frequency is reduced (*Figure 3*). However, in general, one should keep in mind that sampling rates above 40 minutes will decrease the possibility to precisely infer cell divisions and assign daughter cells to their correct mother. Still, the opportunity to reduce laser exposure while maintaining accurate tracking and lineage tracing potential offers a great advantage for long-term imaging experiments of sensitive specimen.

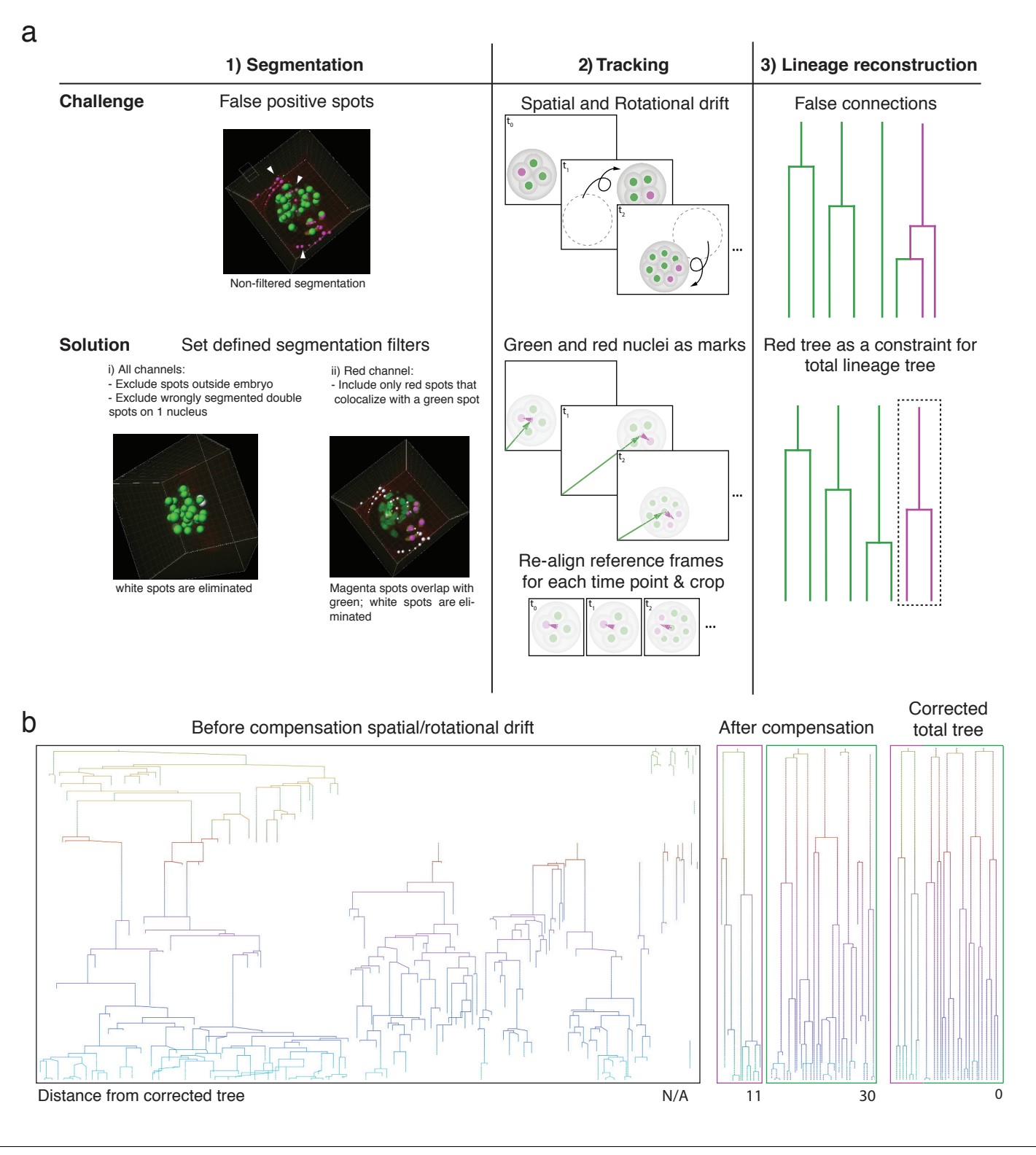

**Figure 2.** Primed track results in efficient lineage reconstruction of embryos with high spatial and rotational drift. (a) Overview of the pipeline used for reliable automated segmentation, tracking, and lineage tracing of the imaged embryos; (1) Segmentation: low thresholds are used for the spot detection in both the green and red channel to enable detection of dimmer cells at later developmental time points. Incorrectly segmented spots are excluded by defined filters: (i) exclusion of spots outside of a defined radius of the embryo, (ii) replacement of incorrectly segmented double spots by one spot per one nucleus, and (ii) exclusion of red spots that do not colocalize with green nuclear spots. (2) Tracking: Spatial drift as well as rapid

*Figure 2 continued on next page*

*Figure 2 continued*

embryo rotation complicates tracking nuclei over prolonged time windows. The segmented nuclei are used for defining reference frames based on the center of mass of the green nuclei and the orientation of the red nuclei. The alignment of the references frames of each time point compensates the spatial and rotational drifts. (3) Lineage tracing: Automated lineage tree reconstruction can make false connections when cells are dividing. By separating the calculation of the lineage trees in the photoconverted red channel from the green channel, the less complex datasets for each channel result in more consistent lineage tracing. pr-mEosFP fluorescence (green) and primed converted pr-mEosFP fluorescence (magenta) overlaid with segmentation results (green and Magenta spheres); Scale bar, 20 µm (b) Lineage trees from the same embryo (corresponding to *Videos 1* and *4*) reconstructed from segmented nuclei before correction for rotational and translational drift (left), after correction for rotational and translational drift for the red channel (second left), after correction for rotational and translational drift for the green channel minus the spots that colocalize with the red spots (second right), and after final manual lineage reconstruction (right). The embryo was imaged every 15 min.

DOI: https://doi.org/10.7554/eLife.44491.009

The following source data and figure supplements are available for figure 2:

**Source data 1.** Summary of challenges with state-of-the-art segmentation and tracking tools.
DOI: https://doi.org/10.7554/eLife.44491.013
**Figure supplement 1.** Dual labeling facilitates segmentation in dense environments.
DOI: https://doi.org/10.7554/eLife.44491.010
**Figure supplement 2.** Embryo dataset size before and after registration.
DOI: https://doi.org/10.7554/eLife.44491.011
**Figure supplement 3.** Comparison of lineage tracing results.
DOI: https://doi.org/10.7554/eLife.44491.012

## Summary and conclusion

In summary, primed Track enables fast, automated, high fidelity lineage tracing of mammalian pre-implantation development combined with reduced illumination time and data volume, key considerations for handling and analyzing data by the biological community (*Pantazis and Supatto, 2014*). A recently published study presents a compelling image analysis framework that enables the long-term tracking of cells during gastrulation and early organogenesis in the post-implantation embryo (*McDole et al., 2018*). Primed Track complements such efforts by enabling accurate fate mapping of mouse pre-implantation embryos.

The ability to correct for both spatial and rotational drift overcomes the previous requirement to exclude spinning embryos from the analysis using primed Track. Furthermore, primed conversion of photoconvertible proteins in combination with primed Track enables the experimenter to still achieve reasonable lineage tracing quality with datasets acquired at lower sampling rate.

The timescales and intensities at which the fluorescent signal of photoconvertible proteins can be observed depend on the expression system (i.e. stable vs. transient expression, promotor choice) as well as the stability of the fusion protein. While we present tracking and lineage tracing of embryos labeled with a relatively highly-expressed and stable H2B-pr-pcFP fusion protein, it is important to take into consideration that low abundant protein fusions may require higher illumination power for visualization, potentially impacting sample integrity. Such cases will in particular benefit from our primed Track pipeline, as it facilitates imaging with longer time intervals while preserving high fidelity in cell tracking and lineage tracing.

In the future, implementing primed conversion to take place inside a SPIM used for volumetric imaging will allow for repeated manual or automatic primed conversion of nuclei once the red fluorescence signal intensity drops below a user-defined threshold. Such pulse-chase experiments can then be extended even longer, ultimately being only limited by the rate of new

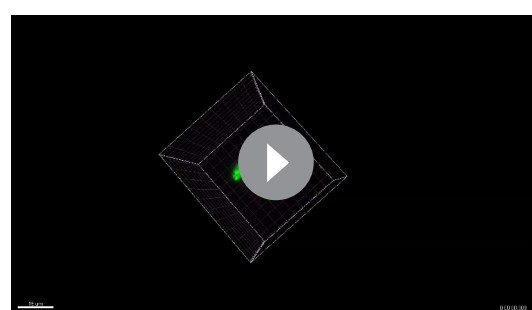

**Video 4.** Video of a developing embryo (same as in *Video 1*) after drift correction. Timelapse video of the example embryo from *Video 1* after drift correction. pr-mEosFP fluorescence (green) and primed converted pr-mEosFP fluorescence (red). Corresponding lineage trees are displayed in *Figure 2d*. Scale bars, 15 µm; framerate: one frame every 15 min.
DOI: https://doi.org/10.7554/eLife.44491.016

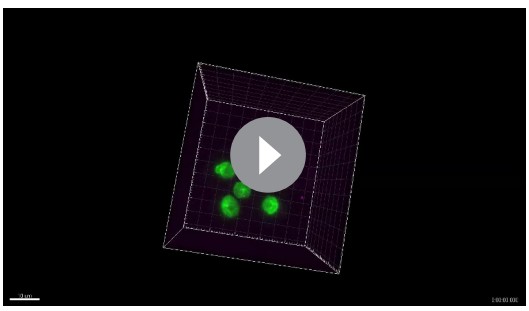

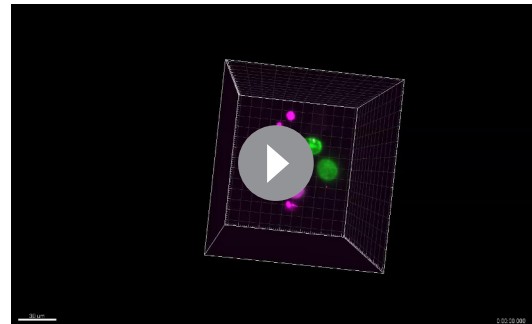

**Video 5.** Video of a developing embryo (same as in *Video 2*) after drift correction. Timelapse video of the example embryo from *Video 2* after drift correction. pr-mEosFP fluorescence (green) and primed converted pr-mEosFP fluorescence (red). Corresponding lineage trees are displayed in *Figure 2—source data 1*. Scale bars, 10 µm; framerate: one frame every 7.5 min.
DOI: https://doi.org/10.7554/eLife.44491.017

**Video 6.** Video of a developing embryo (same as in *Video 3*) after drift correction. Timelapse video of the example embryo from *Video 2* after drift correction. pr-mEosFP fluorescence (green) and primed converted pr-mEosFP fluorescence (red). Corresponding lineage trees are displayed in *Figure 2—source data 1*. Scale bars, 30 µm; framerate: one frame every 5 min.
DOI: https://doi.org/10.7554/eLife.44491.018

green pr-pcFP synthesis. The combination of confined primed conversion of pr-pcFPs with primed Track will allow researchers to get more accurate insight into the dynamic processes responsible for cell fate decisions in the early mammalian embryo.

## Materials and methods

### Molecular cloning and mRNA preparation
The coding sequences for pr-mEosFP and pCS2+-H2B-pr-EosFP were obtained by PCR amplification from pQE32-pr-mEosFP (Addgene No. 99213) and pRSET-pr-mEos2 (gift from Dominique Bourgeois) and cloned into pCS2+-H2B-Dendra2 using AgeI and SnaBI, hence replacing the Dendra2 coding sequence to obtain pCS2+-H2B-pr-EosFP and pCS2+-H2B-pr-Eos2. mRNA was synthesized using the mMESSAGE mMACHINE kit (ThermoFisher Scientific), followed by poly-A-tailing (ThermoFisher Scientific), and purified using a Qiagen RNAeasy kit according to manufacturer guidelines.

### mRNA microinjection of mouse preimplantation embryos and ex utero culture up to four-cell stage
C57Bl/6 wild-type females (Janvier Labs, France) were superovulated by hormone priming, mated to C57Bl/6 males (RRID:IMSR_JAX:000664), and mated females were euthanized by $CO_2$ asphyxiation. Embryos were recovered by flushing oviducts as described previously (*Mohr et al., 2016*; *Plachta et al., 2011*). Embryos were cultured at 37°C and 5% CO2 in KSOM + AA medium covered with mineral oil. mRNA constructs were microinjected into the pro-nucleus at 50 ng/µl or in both cells in two-cell stage embryos, following standard protocols. All these experiments were approved by the veterinary authority of the canton Basel Stadt, Switzerland.

### Confined primed conversion of single nuclei in mouse embryos
Confined primed conversion of single nuclei was performed on mouse embryos at the four-cell stage as previously described in great detail (*Mohr et al., 2016*).

### Volumetric imaging of mouse pre-implantation embryos
Right after confined primed conversion was performed, the four-cell stage embryos were transferred to a pre-equilibrated, custom built inverted SPIM setup suitable for long term imaging of mouse embryos and continuously cultured/imaged until they reached blastocyst stage. For each embryo, a z-stack consisting of 80 planes, 3 µm apart, was acquired every 5, 7.5 or 15 min.

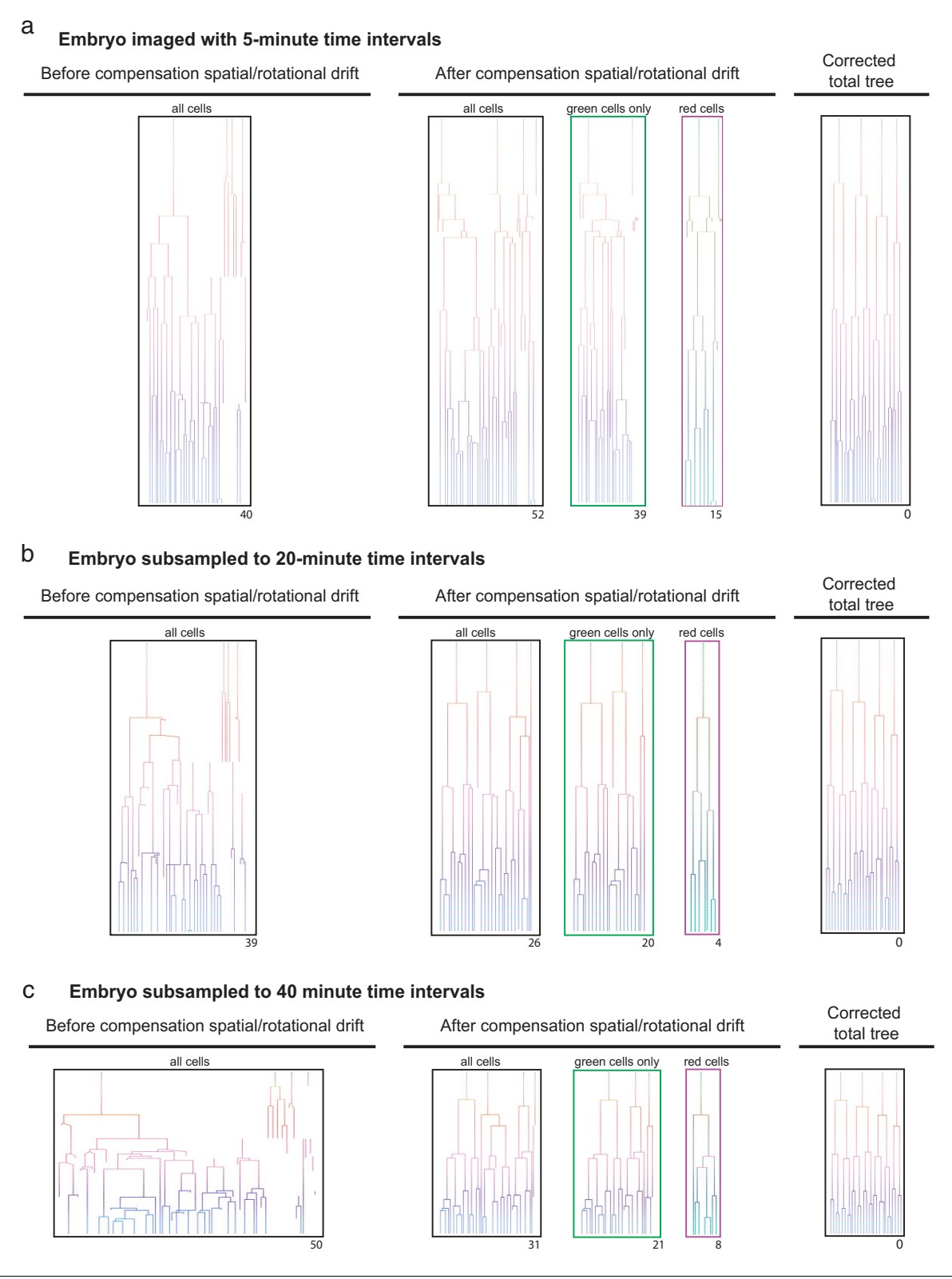

**Figure 3.** Efficient lineage reconstruction using primed Track is still achieved at large imaging time intervals An originally non-rotating embryo imaged with 5 min. (**a**) time intervals were subsampled to 20 min (**b**) and 40 min (**c**) time intervals. Lineage tracing was performed on the non-processed dataset as well as after correction for spatial and rotational drift using the presented approach. The numbers displayed below each lineage tree indicate the distance to the final correct lineage tree.

*Figure 3 continued on next page*

*Figure 3 continued*

DOI: https://doi.org/10.7554/eLife.44491.014

The following figure supplement is available for figure 3:

**Figure supplement 1.** Efficient lineage reconstruction using primed Track is still achieved at large imaging time intervals in a rotating embryo.

DOI: https://doi.org/10.7554/eLife.44491.015

## Mouse embryo lineage tracing

To establish a reference, mouse embryos were lineage traced using the state-of-the-art Imaris lineage tracing package (Bitplane, CH). The automated high-fidelity mouse embryo drift correction and lineage-tracing algorithm described here is explained in detail below.

## Detailed description of primed Track

5D movies of photoconverted mouse embryos were processed with the following pipeline using a custom MATLAB code implemented in Imaris (Bitplane, CH). All codes of primed Track can be downloaded from this code repository: https://git.bsse.ethz.ch/scu_public/primed_track (copy archived at https://github.com/elifesciences-publications/primed_track) (*Ponti, 2018*).

### Cell segmentation

1.Detect green and red cells using the Spot detector in Imaris. Use low threshold to segment all cells even at the cost of including spurious spots. Allow spot radius to be adapted to more accurately fit the volume of the segmented cell. Bright but very small spots can easily be filtered out during segmentation.

### First validation

1. Use the green spot positions to estimate the embryo diameter and discard green spots that are likely to be outside of the embryo. The radius of the embryo is roughly estimated as the median of all maximal inter-spot distances. A user-defined multiplicative factor can optionally compensate for estimation errors and prevent cells at the boundary of the embryo to be discarded should this constraint be too stringent.
2. Search for spots that occur within a small defined distance from a spot in the same channel, discard all wrongly segmented double spots on one nucleus and replace them by one new spot.
3. Discard all spurious red spots that do not colocalize with a green spot. Note that due to the equilibrium between protonated and de-protonated chromophore, green to red photoconversion of pcFPs is never exhaustive and will always retain a green population, rendering this quality control step possible.
4. A red spot discarded during the first validation can optionally be recovered if there is a valid red spot in previous time point within a user-defined search radius. This adjustment compensates for remaining miss-segmentations in the green channel.

### Embryo alignment (drift and rotational correction)/Cropping

1. Imaris Reference Frame Objects are created in MATLAB for each time point: their origin is set at the position of the center of mass (COM) of the green spots and their orientation is given by the vector $\Delta COM = COM^{red} - COM^{green}$. This correction still has one degree of freedom. The rotation angle around the reference frame axis is obtained by comparing the positions of the green spots at timepoints $t$ and $t - 1$ over 360 1-degree rotations and by choosing the angle that minimizes the cell drift between time points. The resampling is performed in Imaris.
2. Crop data to the smallest bounding volume.

### Second validation and subsetting

1. To pick up red cells that were not recovered previously, re-run the first validation on the re-aligned embryo.
2. Create a new spot object that contains the subset of green cells that do not colocalize with red cells.

### Lineage tree reconstruction

1. Imaris' Lineage module is used to track the cells over time and reconstruct their lineage tree. The subsetting in the previous step allows us to reduce the complexity of the lineage tracing problem by breaking it down into two simpler, computationally less expansive, disjoint problems.

## Comparative analysis of lineage trees

To assess the power of our newly created algorithm, we sought to compare the lineage trees obtained with i) no corrections, ii) after embryo realignment with all algorithmic corrections, and iii) after final manual review. We quantified the effects of the corrections and validations on the quality of the lineage trees by calculating the total distance between the lineage trees using the implementation of the tree Zhang-Shasha edit distance algorithm (*Zhang and Shasha, 1989*) by Tim Henderson and Steve Johnson (*Henderson and Johnson, 2013*). The zss algorithm assigns a (user-defined) cost for each node insertion, removal, and update necessary to transform an ordered tree into another, and gives therefore a quantitative measure of dissimilarity of the two trees. Small tracking differences between corrected and uncorrected trees, however, can result in quite large tree distances if the zss algorithm is applied to the complete trees. A correction that relinks one cell to its mother cell in just one time point causes the whole branch to be flagged as incorrect, and the longer the branch, the higher the distance between the trees. In other words, the earlier the tracking error occurs, the larger the distance; yet, only the first time point in the track is incorrect, and its penalty should be the same whether it happens at the beginning of the time series or the end.

To circumvent these issues, we applied the algorithm to a condensed version of the lineage trees. The condensed tree retains only the branch points of the original lineage tree (i.e. the cell divisions). Also, each branch point stores information about the original number of child nodes in its branches (i.e. the number of time points the daughter cells were tracked until their next cell division). The distance between condensed trees will flag positions where cell divisions were tracked incorrectly and tracks that have different lengths, without causing an explosion in the reported distance.

Since our acquisitions started at the four-cell stage, we aimed to build a tree for each of the original four cells (one containing the progeny of the primed converted cell). The final, manually curated lineage was used as ground truth to quantify the effects of the various algorithmic correction steps. The sets of trees across correction schemes were assigned to each other by minimizing the spatial and temporal distance of their origins. After condensation, their pairwise distances were calculated. All distances were summed to give the total lineage tree difference. In addition, spurious trees that resulted from bad segmentation and tracking were not used for the distance calculation, since they already indirectly affected the difference of the tree from which they were erroneously detached.

## Acknowledgements

We thank all members of the Pantazis lab and especially M Haffner for discussion and advice. We thank U Nienhaus and K Nienhaus for discussions and advice as well as E Schreiter, and L Looger for discussions; WP Dempsey for feedback on the manuscript and C Morkunas for administrative management.

## Additional information

### Competing interests

Manuel Alexander Mohr: Is an inventor on a provisional patent application filed by HHMI and ETH Zurich that describes pr-mEosFP. Andrea Boni: Is affiliated with Viventis Microscopy Sàrl. The author has no other competing interests to declare. Periklis Pantazis: Is an inventor on a patent application filed by ETH Zurich and Caltech that describes primed conversion. Is an inventor on a provisional patent application filed by HHMI and ETH Zurich that describes pr-mEosFP. The other authors declare that no competing interests exist.

## Funding

| Funder | Grant reference number | Author |
| --- | --- | --- |
| Nederlandse Organisatie voor Wetenschappelijk Onderzoek | | Maaike Welling |
| Peter und Traudl Engelhorn Stiftung | | Maaike Welling |
| Howard Hughes Medical Institute | | Manuel Alexander Mohr Periklis Pantazis |
| Swiss National Science Foundation | POOP3_157531 | Prisca Liberali |
| European Research Council | ERC-StG-758617 | Prisca Liberali |
| European Research Council | ERC-AdG-695288 | Antoine HFM Peters |
| Swiss National Science Foundation | 31003A_144048 | Periklis Pantazis |
| European Union Seventh Framework Programme | CIG-334552-SIEAVD | Periklis Pantazis |
| Royal Society | Wolfson Research Merit Award | Periklis Pantazis |

The funders had no role in study design, data collection and interpretation, or the decision to submit the work for publication.

### Author contributions
Maaike Welling, Conceptualization, Data curation, Formal analysis, Visualization, Methodology, Writing—original draft; Manuel Alexander Mohr, Conceptualization, Formal analysis, Methodology, Writing—original draft; Aaron Ponti, Software, Writing—original draft; Lluc Rullan Sabater, Data curation, Writing—review and editing; Andrea Boni, Yumiko K Kawamura, Prisca Liberali, Antoine HFM Peters, Pawel Pelczar, Resources, Writing—review and editing; Periklis Pantazis, Conceptualization, Supervision, Funding acquisition, Writing—original draft, Project administration

### Author ORCIDs
Maaike Welling http://orcid.org/0000-0003-3253-3410
Manuel Alexander Mohr https://orcid.org/0000-0002-5189-541X
Aaron Ponti http://orcid.org/0000-0003-4406-3508
Prisca Liberali http://orcid.org/0000-0003-0695-6081
Antoine HFM Peters https://orcid.org/0000-0002-0311-1887
Periklis Pantazis http://orcid.org/0000-0002-8367-9332

### Ethics
Animal experimentation: All these experiments were approved by the veterinary authority of the canton Basel Stadt, Switzerland (Permit Number: 2561).

### Decision letter and Author response
Decision letter https://doi.org/10.7554/eLife.44491.021
Author response https://doi.org/10.7554/eLife.44491.022

# Additional files

### Supplementary files
• Transparent reporting form
DOI: https://doi.org/10.7554/eLife.44491.019

## Data availability

All codes of primed Track can be downloaded from this code repository: https://git.bsse.ethz.ch/scu_public/primed_track (copy archived at https://github.com/elifesciences-publications/primed_track).

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
