## [Decision Letter]

[Editors’ note: a previous version of this study was rejected after peer review, but the authors submitted for reconsideration. The first decision letter after peer review is shown below.]

Thank you for submitting your manuscript "High fidelity lineage tracing in mouse pre-implantation embryos using primed conversion of photoconvertible proteins" for consideration by *eLife*. Your article has been reviewed by three experts and the evaluation has been overseen by a Senior and Reviewing Editor. While the reviewers found the work interesting, the number of substantive questions raised was such that we feel we must reject it. We hope that the reviewer's comments below will be useful to you in revising the manuscript for submission elsewhere. We apologize for not being able to deliver better news, and we hope that you will continue to consider *eLife* for future submissions.

Reviewer #1:

In their manuscript, Pantazis and colleagues demonstrate a combined optical/computational method for reducing the effects of translational and rotational drift in pre-implantation mouse embryo lineage recordings. Primed conversion is used to introduce a sparse second color (red fluorescent nuclei), which is used as a fiducial to reduce drift and an additional quality check on the derived lineages. Although the authors convincingly demonstrate that their method does reduce the effects of drift and thus computational error in their experiments, I am unconvinced that their method is either necessary or generally important for this particular biological application.

While the problem the authors address is a real one, their method does not appear a significant improvement over previous work – in particular the groundbreaking method of Lars Hufnagel and Jan Ellenberg (Strnad et al., 2016). In that manuscript, Hufnagel and Ellenberg performed similar recordings at higher spatiotemporal resolution than reported here. In particular, the temporal sampling in Hufangel and Ellenberg was performed every 5 minutes, 1.5-3x faster than the 7.5 minute and 15 minute recordings performed by Pantazis. One has to wonder if the increased temporal sampling is in fact the dominant source of error in reconstructing lineages – if Pantazis et al. had simply recorded faster, would they have encountered the same degree of drift/error? Hufnagel and Ellenberg claimed a 100% tracking accuracy in their manuscript (for the embryos they ultimately select for lineage analysis) – if this is really the case, I have to wonder why Pantazis et al. did not simply adopt the previous tracking approach. Pantazis et al. compare their computational pipeline to Bitplane Imaris, but the real state-of-the-art comparison is to Hufnagel and Ellenberg. How does the new tracking pipeline presented by Pantazis compare to the coherent point drift method described in this previous work? It is also never explicitly spelled out how many datasets the new method 'rescues', i.e. of the 5/11 embryos that exhibited severe translation/rotational drift, how many were 'recoverable' in the new method? What is the fraction of embryos that are now fully trackable? Because the authors of the current manuscript have failed to put their method into context against previous work, it is difficult to properly assess the impact of their method.

The authors also assert that their method might allow less dosing of the sample (presumably due to the worsened temporal sampling they report) and that their pipeline results in smaller datasets due to the tighter cropping that results. Neither assertion is particularly compelling – (i) I am not convinced that in fact lowering the temporal resolution is advantageous as it seems this makes the tracking problem harder; (ii) the original data sizes they report of ~5GB are hardly massive by today's standards. In summary, I am concerned that the authors' paper constitutes a kind of 'straw man', i.e. they are attacking a problem that has been satisfactorily addressed by previous work. A thorough, statistical comparison of their method to Hufnagel and Ellenberg's would go a long way to convincing me of the value of their method.

Other comments:

The authors are to be commended for comparing H2B-pr-mEosFP to H2B-pr-mEos2. However, I would like to see more evidence for their assertion that their photoconverted embryos develop normally, especially since the primed conversion operation itself intrinsically introduces additional dose. In the previous work by Hufnagel and Ellenberg, 'the tracked embryos had a division timing and number of ICM cells comparable to those of in vitro-cultured embryos… and healthy pups were born after transfer of the imaged embryos into pseudopregnant females…'. Were similar controls done here? What is the additional dose introduced by the primed conversion on the confocal microscope, relative to the light sheet illumination dose used for imaging? The authors image from 4 cell to blastocyst, yet it seems that in previous work it is possible to image from zygote onwards. Is the 4 cell stage necessary due to the increased light sensitivity at earlier stages?

Reviewer #2:

Welling et al. present a combined reverse genetic/optical and computational approach to extract developmental lineages from pre-implantation mouse embryos. The genetic trick relies on photo-convertible proteins that are converted on a confocal set-up and later imaged using light sheet. The computational pipeline extending Imaris achieves proper segmentation, image alignment and uses the total and photo-converted nuclei to improve unsupervised lineaging.

This work has potential, however, for me, it falls short of being a minimal publishable unit. The photoconversion approach has already been published by the authors. What remains is a useful technique that would however fit better into Materials and methods section of a paper focusing on the biology that can be done with this approach. I see the benefits of being able to use the rotating embryos previously excluded from similar analysis (Strnad et al.). However, that is a very niche problem and the pipeline lacks general applicability. The segmentation enhancement is completely dependent on the precise experiment described here, no new algorithm has been presented. Similarly, the re-orientation of the rotating embryos is done using very basic core functions of Imaris. The authors do show that it benefits the analysis of their specific data, however I doubt it will be generally applicable. The comparison of the performance of the Imaris tracker applied to uncorrected and corrected data is a straw man comparison. The Imaris tracker was not developed for tracking lineages in embryos that are fast rotating and therefore it, of course, fails spectacularly.

In order to make the paper work as a methods paper, it would have to be significantly expanded. On the hardware side, the photoconversion would need to happen at one microscope (something the authors clearly intend to do). On the software side, the tracker would need to be benchmarked against existing state of the art tracking solutions such as Ilastik, TrackMate and the Keller pipeline. In addition, the authors would need to show that it is also applicable to other lineaging problems.

Last but not least, the submission contains no code. There is insufficient details provided to reproduce the work, even inside such user friendly software as Imaris is. There is a mention of some MATLAB code that is stringing together the Imaris functionality. At least that needs to be put on github to make this work useful for others. In the current form, it has no impact.

Reviewer #3:

In the short paper entitled "High fidelity lineage tracing in mouse pre-implantation embryos using primed conversion of photoconvertible proteins" the authors use photoconversion of an EosFP by 'primed conversion' to follow by 3D SPIM imaging the cell lineage. In this very limited example the authors propose a potentially promising way of tracking cell fate. However I believe that it currently has a number of issues that should be addressed.

1) Novelty. The novelty here is only mediocre. The photoconversion of EosFP by a 488→730nm illumination pulse has been reported (Mohr, Argast and Pantazis, 2016). Similarly, lineage tracking has been done before (Kurotaki et al., 2007 and others). The novelty is using SPIM here for longer-term tracking, but unfortunately while the potential was there the illumination for both channels was done with the same objective (see point #2).

2) Implementation. The real power of this method should be to focally limit which cells, or region thereof, is getting photoconverted, by launching the light through objectives situated at 90 degrees. Unfortunately, the authors choose to illuminate/activate the cell through only a single objective and thus lose a potential major benefit of the technique. It would have been really neat, and more powerful, to do the activation at a later stage when it would be otherwise difficult to activate only a single cell. In my opinion, doing the activation by cross-beams and in a condition that would be impossible to achieve by a single beam is essential here, and would improve the novelty. The authors ironically discuss axial confinement of the dual activation yet fail to do so and exploit it in the experiments. This must be done.

3) Robustness of the data. It is unclear how many times this experiment was performed. Only once? To show that the technique is robust, more experiments are needed, with statistics. The authors mention that the photoconverted embryos were healthy, but from how many experiments?

4) Other reporters. The authors should show the technique for other reporters, such as in the cell cytosol, or membrane, to generalize the concept.

5) Ambiguity of assignment. It is unclear how long a single lineage can be tracked. The S/N seemed to be high at the later stages. Can the authors better quantify showing the accuracy of assignment in each stage, with statistics.

---

## [Author Response]

[Editors’ note: the author responses to the first round of peer review follow.]

To address the reviewers’ concerns, we performed a set of additional experiments. Specifically, we compared our image analysis pipeline, now referred to as “primed Track”, with three state-of-the-art existing algorithms for cell tracking and lineage tracing. We now show that these algorithms perform poorly in recovering rotating embryos, whereas primed Track faithfully returns high fidelity lineage trees. Primed Track is unique in its ability to perform a comprehensive cell lineage analysis by including rotating embryos, which had to be excluded in previous studies. While we intended to also test the full code developed by Strnad et al., 2016, we were not able to retrieve the full code necessary to repeat their work. We strongly believe that the current comparison to three other state-of-the-art algorithms is sufficient to demonstrate the marked potential of primed Track. We now also show that primed Track is not limited to recovering drifting embryos but also has general value for long-term imaging of sensitive specimen, since primed Track retains the ability for high fidelity tracking and lineage tracing when imaging time intervals are significantly increased (i.e. from 5min to 40min). The opportunity to reduce laser exposure offers a unique advantage for long-term imaging experiments of sensitive specimen, in which single cell tracking and lineage tracing would otherwise not be possible. These additional results demonstrate that primed Track provides superior fidelity in tracking and lineage tracing of drifting specimen over existing algorithms and further allows for increased imaging time intervals, reducing detrimental effects of phototoxicity. We believe that our revised manuscript will be of large interest to the readers of *eLife* with a particular appeal to the volumetric imaging and bioimage informatics community.

Reviewer #1:In their manuscript, Pantazis and colleagues demonstrate a combined optical/computational method for reducing the effects of translational and rotational drift in pre-implantation mouse embryo lineage recordings. Primed conversion is used to introduce a sparse second color (red fluorescent nuclei), which is used as a fiducial to reduce drift and an additional quality check on the derived lineages. Although the authors convincingly demonstrate that their method does reduce the effects of drift and thus computational error in their experiments, I am unconvinced that their method is either necessary or generally important for this particular biological application.While the problem the authors address is a real one, their method does not appear a significant improvement over previous work – in particular the groundbreaking method of Lars Hufnagel and Jan Ellenberg (Strnad et al., 2016). In that manuscript, Hufnagel and Ellenberg performed similar recordings at higher spatiotemporal resolution than reported here. In particular, the temporal sampling in Hufangel and Ellenberg was performed every 5 minutes, 1.5-3x faster than the 7.5 minute and 15 minute recordings performed by Pantazis. One has to wonder if the increased temporal sampling is in fact the dominant source of error in reconstructing lineages – if Pantazis et al. had simply recorded faster, would they have encountered the same degree of drift/error?

We would like to thank reviewer #1 for pointing out this concern. Increased sampling should indeed result in increased lineage tracing fidelity. To test if increased sampling resolves the observed spatial and rotational drift, we performed two sets of light-sheet experiments with 5-minute time intervals. We found that the number of embryos that showed a dramatic spatial/rotational drift did not change when the imaging time intervals are shortened. In fact, the percentage of embryos showing significant rotational and spatial drift when imaged with 5-minute time intervals (4/8 embryos) was similar to the embryos imaged with 7.5-minute or 15-minute time intervals (5/11 embryos).

Hufnagel and Ellenberg claimed a 100% tracking accuracy in their manuscript (for the embryos they ultimately select for lineage analysis) – if this is really the case, I have to wonder why Pantazis et al. did not simply adopt the previous tracking approach. Pantazis et al. compare their computational pipeline to Bitplane Imaris, but the real state-of-the-art comparison is to Hufnagel and Ellenberg. How does the new tracking pipeline presented by Pantazis compare to the coherent point drift method described in this previous work?

Ellenberg and colleagues indeed published an elegant two-step segmentation and tracking pipeline that reliably reconstructed lineages from pre-implantation embryos. However, this method was developed for embryos that were imaged at 5-minute time intervals and that did not display rotational or spatial drift. It is worth noting that Ellenberg and colleagues reported that they excluded a significant fraction of embryos (n = 6 out of 19 embryos) that ‘[…] rotated too rapidly for automatic alignment’ (Strnad et al., 2016). In addition, Ellenberg and colleagues reported that even lineage trees reconstructions from non-rotating embryos required manual corrections.

In our manuscript, we show that we are able to reliably reconstruct lineages from embryos that display heavy spatial and rotational drift using an image analysis pipeline (referred to as “primed Track”) that takes advantage of the dual labeling in the embryos, an approach that would otherwise not have been possible. Importantly, we now show that primed Track even allows for the reliable reconstruction of lineages from embryos imaged with larger time intervals. This achievement provides unique opportunities for long-term imaging of sensitive and/or dim specimen that require higher laser intensities for visualization of fluorescent signals while limiting potential effects on developmental progression.

It is also never explicitly spelled out how many datasets the new method 'rescues', i.e. of the 5/11 embryos that exhibited severe translation/rotational drift, how many were 'recoverable' in the new method? What is the fraction of embryos that are now fully trackable?

We were able to recover 100% of the rotating embryos. This accomplishment is due to the fact that embryos were labeled with an optimized photoconvertible protein, pr-mEosFP. It provided sufficient contrast for robust labeling up to the blastocyst stage due to its superior brightness and photostability when compared to previously employed fluorescent protein versions. Consequently, primed Track correctly segmented all labeled nuclei and realigned the embryos, allowing for the reliable reconstruction of the lineage trees. In both realigned embryos and non-rotating embryos, occasional wrong daughter cell assignment after cell division required limited manual correction to obtain maximal fidelity.

Because the authors of the current manuscript have failed to put their method into context against previous work, it is difficult to properly assess the impact of their method.

To compare our work to other existing state-of-the-art lineage tracing tools, we tested the performance of the Ilastik, the TrackMate and the TGMM algorithm for segmentation and lineage tracing developed by the Keller lab on our data. We intended to also test the algorithm developed by Ellenberg and colleagues, yet the code provided in the published paper turned out to be incomplete; the authors were notified of this discrepancy.

We found that all sophisticated segmentation and tracking tools were not able to reconstruct correct lineage trees from rotating embryos. In pre-implantation embryos, daughter cells experience dramatic repositioning relative to their division plane. Due to these dynamics upon cell division, Ilastik, TrackMate and TGMM had difficulties in correctly assigning daughter cells to their respective mother cells after cell division. Combined with the substantial spatial and rotational drift of pre-implantation embryos, tracking of cells became uncertain for tested lineage tracing tools. In addition, both Ilastik and TrackMate showed limitations in the correct segmentation of individual cells when the cell density in the embryo increased.

In primed Track we overcame these limitations: the Imaris spot detection tool was enhanced with additional customizable parameters that disabled the segmentation of background signal (i.e. signal outside a certain radius of the embryos) and that required the overlap of segmented red spots with green spots in order to distinguish real nuclear signal from falsely segmented background signal. A concise summary of the main limitations of tested algorithms is included in Figure 2—figure supplement 3 in the revised manuscript.

The authors also assert that their method might allow less dosing of the sample (presumably due to the worsened temporal sampling they report) and that their pipeline results in smaller datasets due to the tighter cropping that results. Neither assertion is particularly compelling – (i) I am not convinced that in fact lowering the temporal resolution is advantageous as it seems this makes the tracking problem harder; (ii) the original data sizes they report of ~5GB are hardly massive by today's standards. In summary, I am concerned that the authors' paper constitutes a kind of 'straw man', i.e. they are attacking a problem that has been satisfactorily addressed by previous work. A thorough, statistical comparison of their method to Hufnagel and Ellenberg's would go a long way to convincing me of the value of their method.

We thank reviewer #1 for raising these important points that we would like to address as follows:

1) Mouse pre-implantation embryos are very sensitive to laser illumination. Despite the fact that specimen imaged with a SPIM are only excited in the focal plane, we found that embryos arrest before reaching the blastocyst stage upon higher laser intensities and/or higher sampling intervals. Specifically, we saw that the exact laser intensities that allowed the development of 50% of the embryos to the blastocysts stage at imaging intervals of 7.5 minutes, did not allow any of the embryos (n=10) to develop past the 8-cell to morula stage when sampled every 5 minutes. In order for us to be able to image our embryos with a higher sampling frequency, we had to decrease the power of laser illumination. The possibility to image with larger time intervals is therefore of particular importance for embryos that possess weaker fluorescent signal and need higher laser dosage for detection. Examples include but are not limited to endogenously tagged embryos expressing a fluorescent protein fusion at physiological amounts as opposed to high overexpression using potent promoters. To test the ability to perform lineage tracing in embryos imaged with larger time intervals, we subsampled both rotating and non-rotating embryos and applied primed Track to realign the embryos and reconstruct lineage trees. While we did see a small decrease in lineage tracing fidelity compared to embryos imaged with shorter time intervals, primed Track was able to still reconstruct lineages from both rotating and non-rotating embryos imaged every 30 or 40 minutes, respectively (see Figure 3 and Figure 3—figure supplement 1).

2)The sizes of the original data are dependent on the sampling rate of the embryos as well as on the duration of the time series. Therefore, the size of the original data ranged from 5GB to 25GB. While storage of large data does not pose a challenge these days, the execution speed of fundamental functions of available bioimage analysis software solutions can be severely affected by the size of the dataset. For instance, the need to load and analyze data sets for the optimization of image analysis parameters took several hours for 4D datasets on GPU-accelerated high-performance workstations. Consequently, cropping data imaged with larger time intervals to less than half of the original size greatly decreased the amount of time needed for analysis calculations.

Other comments:The authors are to be commended for comparing H2B-pr-mEosFP to H2B-pr-mEos2. However, I would like to see more evidence for their assertion that their photoconverted embryos develop normally, especially since the primed conversion operation itself intrinsically introduces additional dose. In the previous work by Hufnagel and Ellenberg, 'the tracked embryos had a division timing and number of ICM cells comparable to those of in vitro-cultured embryos… and healthy pups were born after transfer of the imaged embryos into pseudopregnant females…'. Were similar controls done here? What is the additional dose introduced by the primed conversion on the confocal microscope, relative to the light sheet illumination dose used for imaging?

We thank reviewer #1 for pointing out this issue that we now address in detail in the revised manuscript. Before embryos were transferred to the light-sheet microscope for volumetric imaging, they were exposed to 20 seconds of continuous-wave 488nm and 730nm light required for effective confined primed conversion of H2B-pr-mEosFP. Normal development of H2B-pr-mEosFP injected embryos was evaluated by comparing the progression of primed converted and non-converted embryos to the blastocyst stage. Under these imaging conditions, the percentage of embryos that reached the blastocysts stage (see Figure 1—figure supplement 2B) was similar for both groups, indicating that neither confined primed conversion nor expression of H2B-pr-mEosFP affected their development.

The authors image from 4 cell to blastocyst, yet it seems that in previous work it is possible to image from zygote onwards. Is the 4 cell stage necessary due to the increased light sensitivity at earlier stages?

The reason to start imaging at the 4-cell stage was for practical reasons: due to absence of a transgenic line expressing H2B-pr-mEosFP protein, we injected embryos with H2B-prmEosFP mRNA at the zygote or 2-cell stage that rendered H2B-pr-mEosFP protein robustly visible for our lineage tracing approach mostly at the 4-cell stage (i.e. enough red contrast could be generated upon confined primed conversion for lineage tracing).

Reviewer #2:[…] This work has potential, however, for me, it falls short of being a minimal publishable unit. The photoconversion approach has already been published by the authors. What remains is a useful technique that would however fit better into Materials and methods section of a paper focusing on the biology that can be done with this approach. I see the benefits of being able to use the rotating embryos previously excluded from similar analysis (Strnad et al.). However, that is a very niche problem and the pipeline lacks general applicability.

We thank reviewer #2 for the comments and suggestions. In the revised manuscript we expanded on the applicability and features of our image analysis pipeline primed Track. Specifically, we have shown that primed Track is indeed unique in its ability to reliably realign rotating embryos, which is required for faithful reconstruction of lineage trees, a task that cannot be accomplished with currently existing state-of-the-art image analysis algorithms. Using primed Track, a significant amount of valuable early embryos (~45% in our dataset) can be restored and used for further analysis. In addition, we now prove that primed Track is able to reconstruct lineage trees from embryos sampled at a much lower temporal rate. Although the lower sampling density results in an increased displacement of individual cells in consecutive time points, primed Track allows for reliable lineage tree reconstruction of embryos imaged with larger time intervals.

Importantly, the opportunity to image with larger time intervals while maintaining the capacity to perform lineage tracing will have significant beneficial consequences for the developmental imaging community: (i) decreased phototoxicity in specimen that require higher laser intensities for visualization of relatively weak fluorescent signal of e.g. endogenous tagged proteins of interest, and (ii) extended volumetric imaging of sensitive specimen such as embryos and organoids where lineage tracing is an important tool for the understanding of principles of cell fate decision and self-organization, respectively.

The segmentation enhancement is completely dependent on the precise experiment described here, no new algorithm has been presented. Similarly, the re-orientation of the rotating embryos is done using very basic core functions of Imaris. The authors do show that it benefits the analysis of their specific data, however I doubt it will be generally applicable. The comparison of the performance of the Imaris tracker applied to uncorrected and corrected data is a straw man comparison. The Imaris tracker was not developed for tracking lineages in embryos that are fast rotating and therefore it, of course, fails spectacularly.

Like all other currently existing tracking and lineage tracing algorithms, Imaris itself does not provide a function that allows for the reorientation of segmented images and therefore is not able to track our non-corrected datasets. To address this need, we introduced primed Track, a code for calculating the center of mass of both the green channel and the red channel, which allows exact reorientation of drifting embryos. Our data demonstrate that primed Track is unique in enabling precise realignment of rotating embryos and can accomplish high fidelity tracking and lineage tracing in Imaris, essentially outperforming current gold standard algorithms for lineage tracing.

Moreover, we now provide proof that primed Track can accomplish robust lineage tracing of samples imaged with significantly larger time intervals, providing important benefits for volumetric imaging of sensitive specimen.

In order to make the paper work as a methods paper, it would have to be significantly expanded. On the hardware side, the photoconversion would need to happen at one microscope (something the authors clearly intend to do).

The goal of the current study was the development of an integrated image analysis algorithm that allows for high fidelity segmentation, tracking, and lineage tracing of specimen that display substantial spatial and rotational drift. While we are planning the implementation of primed conversion into a light-sheet microscope, we feel that this is out of the scope of the currently presented work. For more detail, please see our answer to point 2 of reviewer #3.

On the software side, the tracker would need to be benchmarked against existing state of the art tracking solutions such as Ilastik, TrackMate and the Keller pipeline.

We tested the performance of Ilastik, TrackMate and the TGMM algorithm for segmentation and lineage tracing developed by the Keller lab on our data and show now that these state-of-the-art tracking and lineage tracing algorithms all fail to reconstruct lineage trees from embryos that display substantial drift (summarized in Figure 2—figure supplement 3). More detailed information is provided in the third comment for reviewer #1.

In addition, the authors would need to show that it is also applicable to other lineaging problems.

Besides showing that our image analysis algorithm can be used to recover embryos that display significant spatial and rotational drift, we now also show that the use of our pipeline to reconstruct lineage trees is beneficial when working with lower sampling frequency. Timelapse imaging with longer time intervals will automatically result in lower quality lineage reconstruction, yet primed Track is able to realign embryos sampled with up to 40-minute time intervals (see Figure 3). The resulting lineage trees are of much higher fidelity than when these embryos are tracked and lineage traced using their original non-corrected data. The ability to reduce laser exposure while maintaining accurate tracking and lineage tracing abilities offers a great advantage for long-term imaging experiments of sensitive specimen.

Last but not least, the submission contains no code. There is insufficient details provided to reproduce the work, even inside such user friendly software as Imaris is. There is a mention of some MATLAB code that is stringing together the Imaris functionality. At least that needs to be put on github to make this work useful for others. In the current form, it has no impact.

To enable immediate dissemination of primed Track, we now provide a link in the manuscript to the entire code used for our bioimage analysis pipeline.

Reviewer #3:In the short paper entitled "High fidelity lineage tracing in mouse pre-implantation embryos using primed conversion of photoconvertible proteins" the authors use photoconversion of an EosFP by 'primed conversion' to follow by 3D SPIM imaging the cell lineage. In this very limited example the authors propose a potentially promising way of tracking cell fate. However I believe that it currently has a number of issues that should be addressed.1) Novelty. The novelty here is only mediocre. The photoconversion of EosFP by a 488→730nm illumination pulse has been reported (Mohr, Argast and Pantazis, 2016). Similarly, lineage tracking has been done before (Kurotaki et al., 2007 and others). The novelty is using SPIM here for longer-term tracking, but unfortunately while the potential was there the illumination for both channels was done with the same objective (see point #2).

While cell tracking and lineage tracing approaches of early mouse embryos in a light-sheet microscope have indeed been shown before, a large proportion of embryos were excluded from analysis due to significant rotational drift (31% in the study from Strnad et al., 45% in our study) limiting developmental studies. Here, we present primed Track, an image analysis pipeline that is able to reliably register photoconverted nuclei and reorient embryos to compensate for dramatic drift. Primed Track takes advantage of previous achievements of the lab: we labeled the nuclei of mouse embryos with a rationally engineered primed convertible protein, pr-mEosFP, which possess improved photostability and brightness, and performed non-toxic, confined primed conversion of one nucleus in a 4-cell stage embryo. Using this strategy, we accomplished both global (green) and sparse (red) labeling of nuclei in the embryos that served as fiducial markers for registration. As a result, we were able to recover lineage trees from all rotating embryos in our analysis, an achievement that is not possible with currently existing tracking and lineage tracing tools.

Moreover, in the revised manuscript we now show that the use of our pipeline to reconstruct lineage trees is beneficial when working with lower sampling frequency. Although time-lapse imaging with longer time intervals typically result in lower quality lineage reconstruction, primed Track can robustly and reliably realign embryos sampled with up to 40-minute time intervals (see Figure 3). The resulting lineage tree fidelity is superior to embryos that were cell tracked and lineage traced with original coordinates. The ability to reduce laser exposure while maintaining accurate tracking and lineage tracing potential offers a great advantage for longterm imaging experiments of sensitive specimen.

2) Implementation. The real power of this method should be to focally limit which cells, or region thereof, is getting photoconverted, by launching the light through objectives situated at 90 degrees. Unfortunately, the authors choose to illuminate/activate the cell through only a single objective and thus lose a potential major benefit of the technique. It would have been really neat, and more powerful, to do the activation at a later stage when it would be otherwise difficult to activate only a single cell. In my opinion, doing the activation by cross-beams and in a condition that would be impossible to achieve by a single beam is essential here, and would improve the novelty. The authors ironically discuss axial confinement of the dual activation yet fail to do so and exploit it in the experiments. This must be done.

Primed Track is able to reliably reconstruct lineage trees of specimen even when these specimen experience substantial drift. In order for primed Track to successfully accomplish this goal, single nuclei need to be precisely photoconverted in the developing mouse embryo.

To accomplish axially confined photoconversion, we took advantage of our previously developed advanced imaging modality ‘primed conversion’ (Dempsey et al., 2015). We achieved confined photoconversion of H2B-pr-mEosFP by axially confining primed conversion through the selective intersection of the priming 488nm and the converting 730nm beam in a common focal spot. To accomplish the laser intersection, we simply added a commercially available primed conversion filter cube before the objective aperture, thereby separating both beams until the focal plane in a commercial confocal laser-scanning microscope (CLSM) as previously described (Mohr et al., 2015). Importantly, this straightforward implementation allows for precise primed conversion of volumes much smaller than one nucleus in more crowded 3D environments.

Hence, effective primed Track does not depend on implementing primed conversion into a SPIM, as sufficient confinement can be accomplished with our approach. While the opportunity to perform both primed conversion and long-term volumetric imaging in the same microscope would be neat for confined photoconversion at later developmental stages, we feel that this request is beyond the scope of the presented bioimage informatics study. Consequently, we included this consideration in the Discussion of the manuscript.

3) Robustness of the data. It is unclear how many times this experiment was performed. Only once? To show that the technique is robust, more experiments are needed, with statistics. The authors mention that the photoconverted embryos were healthy, but from how many experiments?

We acquired time-lapse data from 19 embryos that underwent primed conversion and developed to blastocysts. Out of those embryos 9 displayed significant spatial and rotational drift. We were able to recover 100% of the rotating embryos. We tested the health of the prmEosFP injected embryos by comparing the development of primed converted and nonconverted embryos to the blastocyst stage in vitro. We did not find a difference in the percentage of embryos that reached the blastocysts stage for these two groups (see Figure 1—figure supplement 2B), indicating that primed conversion did not affect their development.

4) Other reporters. The authors should show the technique for other reporters, such as in the cell cytosol, or membrane, to generalize the concept.

We decided to perform our analysis using fluorescent histone labeling, as it is the gold standard for lineage tracing in various model organisms. Histone labeling stays associated with the cell’s DNA and does not diffuse upon cell division, providing precise spatiotemporal orientation of nuclei throughout the cell cycle. As its signal does not overlap with other cells, fluorescent histone labeling is an exclusive marker for individual cells. Furthermore, histones have a very long half-life (reviewed by Toyama et al., 2013), and dilution of contrast is predominantly due to cell division during mouse embryonic development.

We have not used primed Track with a cytoplasmic or a membrane label, because these labeling strategies pose several disadvantages compared to histone labels. For both membrane labeled cells as well as cytoplasmic labeled cells, precise photoconversion of individual cells will be problematic, because it will be difficult to discern the exact cell boundaries of two adjacent cells that are in contact with each other. Furthermore, cell division will cause diffusion of the fluorescent signal which will also suffer from higher turnover rates.

In addition, early embryos have a relatively large cytoplasm which will result in dilution of the fluorescent signal and a decrease in its detectability. Last but not least, the segmentation of individual cells will be much more complicated using cytoplasmic labels.

5) Ambiguity of assignment. It is unclear how long a single lineage can be tracked. The S/N seemed to be high at the later stages. Can the authors better quantify showing the accuracy of assignment in each stage, with statistics.

Ambiguity in segmentation and tracking of individual cells can potentially be caused i) by the increase in cell numbers that are densely packed at later developmental stages, and ii) by the diminishing photoconverted red fluorescent signal that gets diluted upon repeated cell divisions. Here, we were able to reliably segment and precisely track all nuclei up to the blastocyst stage in all embryos that we acquired, because we labelled embryos with an optimized photoconvertible protein, pr-mEosFP. It provided sufficient contrast for robust labeling up to the blastocyst stage due to its superior brightness and photostability when compared to previously employed fluorescent protein versions.

We compared the lineage trees of rotating embryos generated without corrections and using primed Track with manually corrected ground truth trees. We calculated the distance between these trees using a method that gives penalties for incorrect cell divisions and track length. While it was not possible to compare ground truth trees to lineage trees reconstructed from non-corrected datasets due to the lack of tree similarity, we showed that we could reliably reconstruct lineage trees from corrected datasets. Moreover, we were able to simplify the lineage tree reconstruction by separating the calculation of the trees from converted red cells from those of non-converted green cells. The details can be found in the Materials and methods section of our manuscript (in “Comparative analysis of lineage trees”).